# Laser nano-filament explosion for enabling open-grating sensing in optical fibre

Keivan Mahmoud Aghdami [1,2,3✉], Abdullah Rahnama [1,3✉], Erden Ertorer[1] & Peter R. Herman [1]

Embedding strong photonic stopbands into traditional optical fibre that can directly access and sense the outside environment is challenging, relying on tedious nano-processing steps that result in fragile thinned fibre. Ultrashort-pulsed laser filaments have recently provided a non-contact means of opening high-aspect ratio nano-holes inside of bulk transparent glasses. This method has been extended here to optical fibre, resulting in high density arrays of laser filamented holes penetrating transversely through the silica cladding and guiding core to provide high refractive index contrast Bragg gratings in the telecommunication band. The point-by-point fabrication was combined with post-chemical etching to engineer strong photonic stopbands directly inside of the compact and flexible fibre. Fibre Bragg gratings with sharply resolved $\pi$-shifts are presented for high resolution refractive index sensing from $n_H = 1$ to 1.67 as the nano-holes were readily wetted and filled with various solvents and oils through an intact fibre cladding.

[1] Department of Electrical and Computer Engineering, University of Toronto, 10 King's College Rd., Toronto, ON M5S 3G4, Canada. [2] Department of Physics, Payame Noor University (PNU), P.O. Box: 19395-4697 Tehran, Iran. [3] These authors contributed equally: Keivan Mahmoud Aghdami, Abdullah Rahnama. ✉email: k_aghdami@pnu.ac.ir; abdullah.rahnama@mail.utoronto.ca

The compact and flexible format of optical fibre serves broadly as a high-capacity conduit in today's information highway. These advantages further facilitate optical sensing of the local environment that can reach over short to long distances and probe into challenging environments[1,2]. By the nature of strong optical confinement, standard fibres require micro- or nano-engineering of devices directly inside of the core waveguide that is frequently applied by means of chemical etching[3], mechanical polishing[4], thermal tapering[5,6], laser modification[7,8], or ion milling[9]. Such structures redirect the waveguiding light to probe outside of the fibre cladding. However, optical probing is significantly more responsive when the external environment can be brought close to or directly into the fibre core as provided by photonic bandgap hollow-core fibre. The open structure enables refractive index (RI) sensing of gases[10] and liquids[11,12] but without the advantages for remote, localised or distributed sensing that is otherwise possible when micro-devices such as open cavities, micro-optical resonators and interferometers[13] have been embedded inside of a solid core fibre. Such micro-devices enable localised points of optical sensing at the core waveguide by basic means of absorption, fluorescence, scattering (Rayleigh, Mie, and Brillouin), diffraction, and interferometry[14–18]. These internal elements improve optical sensitivity that enables broadly based lab-in-fibre applications in biology[19] or healthcare[20] such as label-free detection of cancer biomarkers[21].

Fibre Bragg gratings (FBG) are one favoured device in localised optical fibre sensing owing to sharp and environmentally responsive resonances[3]. The FBG further benefits from the ease of external laser fabrication through the cladding core[22]. However, the FBG typically probes the local environment through the cladding, without benefiting from the open photonic bandgap structure such as available in silicon photonic or another planar optical circuit technologies[17,23–25]. An open periodic structure with high RI contrast offers strong photonic bandgap responses, which can be spectrally and spatially narrowed by optical defects to provide high-Q micro-resonators for highly localised sensing. Such micro-resonators enable optical trapping[26], label-free sensing[27] or single nano-particle detection[28] on a chip, but have been limited in fibre by a solid core FBG buried deeply in the cladding. Various means of chemical, laser, and mechanical machining or thinning of the cladding have facilitated evanescent sensing at the surface of the FBG[14,29–31]. An open-structured photonic crystal has otherwise been challenging to fabricate transversely into the fibre core. Ion milling or laser machining at the core of a thinned fibre has provided FBG sensors with nano-structured surface relief[32,33], blind holes[34], and through holes[35,36]. However, the cladding processing renders such fibres mechanically fragile and lacking robustness for practical application. Yang et al.[37] accessed the core waveguide through a narrow cladding channel to open an array of micro-holes by chemical etching of self-focused laser tracks. Third-order Bragg stopbands provided a microfluidic RI-sensing response, but were restricted to the low sensitivity of 5 nm/RIU and narrow RI window from 1.32–1.41 owing to a large hole diameter (1.35 µm) exceeding the optical wavelength.

Femtosecond lasers provide an alternative direction for opening high-aspect-ratio holes in transparent materials by a single-pulse nano-explosion from a filament-shaped beam focus. Hole diameters of ~200 nm diameter have been demonstrated in glasses by Kerr lensing, and axicon optics[38–43]. Our group has further harnessed surface aberration of glass plates to form similarly long filament tracks in bulk glass[44]. This approach was extended into optical fibre by using RI-matching fluid to eliminate astigmatism of the cylindrical fibre shape and generate filament arrays having first-order Bragg stopbands[45–47].

In this work, the laser interaction was further scaled up to drive a controlled filament explosion in the fibre cross-section, resulting in densely packed arrays of uniform nanoholes[48]. The isolated, blind or through holes were patterned with controllable positioning to selectively pierce the core and/or cladding and enabled strong photonic stopbands to be formed directly in the core waveguide with minimal processing of the surrounding cladding. FBGs with strong resonances and sharply resolved π-shifts are presented that provided high capillarity for wetting with various solvents and oils, demonstrating high-resolution RI sensing from $n_H = 1–1.67$ through an intact fibre cladding.

## Results

**Nanohole FBG—modelling stopbands.** A schematic representation of the nanohole array formation by laser filament nano-explosion is depicted in Fig. 1. The approach meets four key challenges for nano-structuring of strong photonic stopbands in optical fibre. First, the beam delivery avoids astigmatic aberration by the cylindrical cladding shape (Fig. 1), resulting in the formation of nanoholes partially to fully through the fibre cross-section, without thinning or inducing deleterious damage in the fibre cladding structure. Second, the nanohole processing facilitates nanohole assembly on small periodicity at optical wavelength scale, without inducing melting or significant heat-affected zone. The resulting 0.46 contrast in RI provides strong photonic bandgap responses without high optical scattering loss. Third, the laser direct writing affords flexible and rapid patterning, to tune the spectral response and provide micro-cavity like responses such as π-shifted FBGs. Chemical etching facilitates photonic bandgap engineering by tuning the nanohole diameter from 200 nm to 700 nm. And fourth, the nanoholes draw significant capillary force to enable wetting by a wide range of solvent types fully through the 125 µm diameter of the fibre (Supplementary Movie 1, 2). The nanohole array thus defines an open and flexible FBG-sensing structure that can be fabricated rapidly in a single-step procedure, and provide robust mechanical integrity together with strong optofluidic responses due to the subwavelength hole diameter.

The anticipated optofluidic responses of the nanohole FBG structure (Fig. 1) are significantly stronger and more sensitive than with the traditional FBG grating as demonstrated by the simulations in Fig. 2. The second-order grating responses were modelled in standard telecommunication fibre (SMF-28) having 300 nm diameter holes with 1072 nm periodicity and a relatively short length of 643 µm. The potential for precise RI sensing is

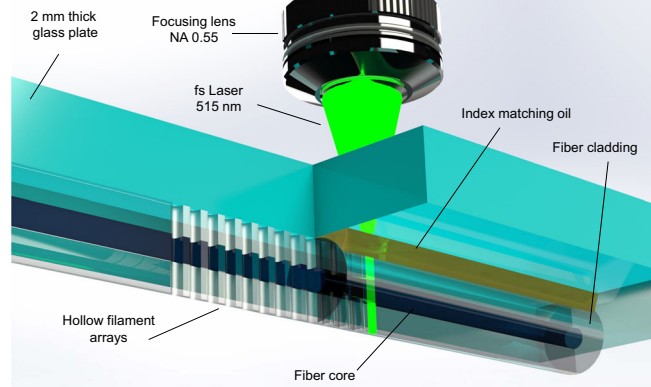

**Fig. 1 Femtosecond laser filament processing in optical fibre.** Schematical arrangement of optical aberration plate and optical fibre used for generating femtosecond laser filaments, and opening high-aspect ratio nanoholes through the fibre cladding and core waveguide.

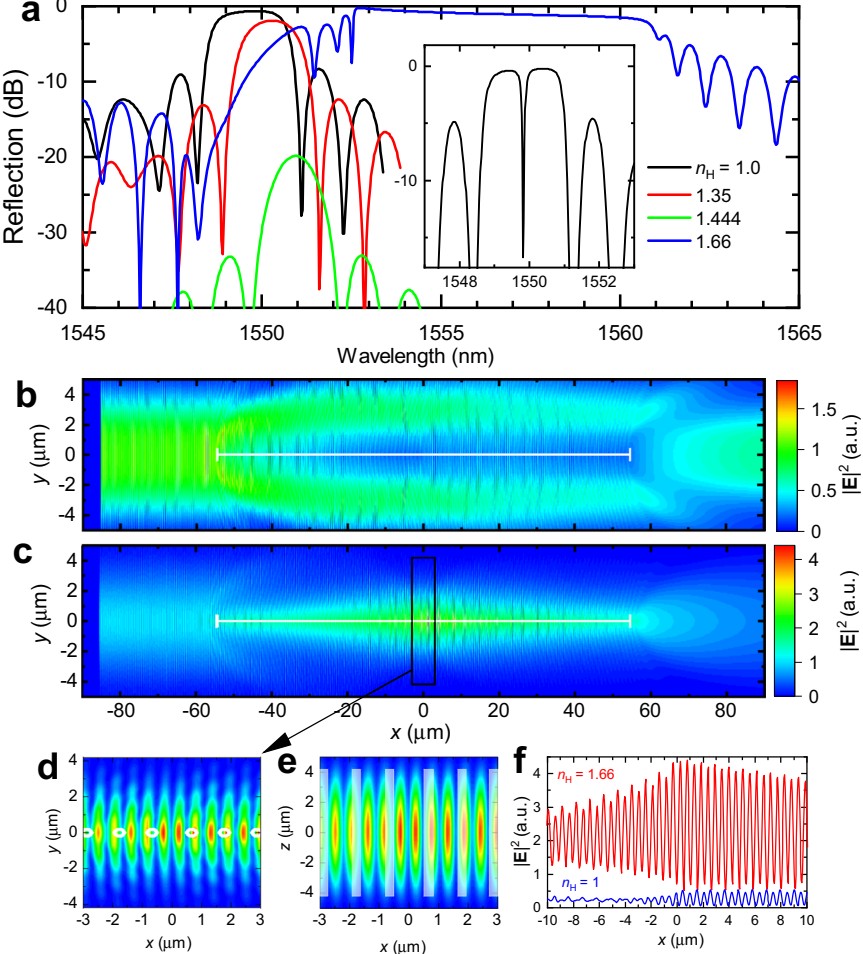

**Fig. 2 Simulated spectra and light-field distribution inside of the nanohole arrays.** Simulated reflection spectra (**a**) for a second-order FBG of uniformly pitched nanoholes ($\Lambda = 1.072\,\mu m$) having 300 nm diameter, 0.643 mm array length (600 holes) and high RI contrast ($\Delta n = -0.45$ to $+0.21$) when filled with air, solvent or oil. A sharply resolved transmission band (30 pm FWHM) has opened (inset spectrum, $n_H = 1$) from a $\pi$-shifted FBG of 1200 holes. The simulated intensity distribution transverse to nanoholes (xy plane) for low ($n_H = 1.0$) (**b**) and high ($n_H = 1.66$) (**c**) RI cases of a $\pi$-shifted FBG (100 holes, $x = -53.6$ to $+53.6\,\mu m$). Magnified views for the high RI case reveal intensity concentration in the filament plane (**e**) and around the $\pi$-defect (**d**; $x = 0$). White circles and bars outline the hole positions. Axial intensity oscillation (x axis, $y = z = 0$) comparing (**f**) high and low RI cases.

noted by the strong and rapidly shifting stopbands (Fig. 2a) when nanoholes are filled with air and solvents having different RI values spanning from $n_H = 1$–1.66 (see also Supplementary Fig. 1). The subwavelength hole diameter provides increasingly stronger (i.e., >−2 dB for air) and broader resonances (up to 7 nm bandwidth), rising up for both increasing or decreasing RI from the matching condition at $n_H \cong n_{eff} = 1.450$ (Fig. 2a). With variable hole diameter, the EME modelling indicated a high RI sensitivity response of up to 600 nm/RIU will be available at high RI (i.e., $n_H = 1.66$) and a large hole diameter of 700 nm.

When imposing a $\pi$-shifted defect into the nanohole array, the simulated stopband (Fig. 2a, inset spectrum) opens a sharp and narrow transmission resonance ($\Delta\lambda = 30$ pm, 3 dB bandwidth) inside of the air-filled stopband, representing a moderately strong resonator quality factor ($Q \sim 10^4$). The strong influence of the $\pi$-defect and nanohole array on the guided light-field distribution is demonstrated by the high repulsion (Fig. 2b) or attraction (Fig. 2c) of the mode within a narrow ($\sim 2.5\,\mu m$) zone following along nanoholes filled with air ($n_H = 1$) or oil ($n_H = 1.66$), respectively. For the higher RI case ($n_H = 1.66$), magnified views of the intensity profiles near the $\pi$-defect ($x = 0$) reveal an asymmetric narrowing from the fundamental mode field from $10.4 \pm 0.5\,\mu m$ diameter to $5\,\mu m$ (Fig. 2d) and $8\,\mu m$ (Fig. 2e) for

respective perpendicular and parallel directions with respect to the hole axis. The beginning of a micro-cavity response is noted along the fibre centre axis (Fig. 2f) by the light intensity drawing into a short $66\,\mu m$ zone (full 3 dB) as presented for the air ($n_H = 1$) and oil ($n_H = 1.66$) cases (Fig. 2f).

**Nanohole array FBG—morphology.** Long and uniform laser filament shapes (Fig. 1) of up to $125\,\mu m$ length were provided by surface aberration from glass plates of up to 3 mm thickness. In this arrangement, high laser pulse energies of $7\,\mu J$ drove a uniform nano-filament explosion without distortion from Kerr lens focusing or plasma defocusing effects. The nonlinear optical benefits of narrowing modification dimensions below diffraction-limited sizes were thus retained. Isolated, blind, and through holes could be tailored by the laser and focussing controls to any position in the fibre cross-section. An example of an FBG with nanohole arrays reaching fully through the cladding and core is presented in Fig. 3. A schematic of the FBG (centre image) represents the dense packing of the nanoholes on $1.072\,\mu m$ periodicity, targeting a high-aspect ratio of processing control reaching through the 125 nm diameter fibre cladding. The evidence for this high-aspect-ratio hole geometry is provided in the

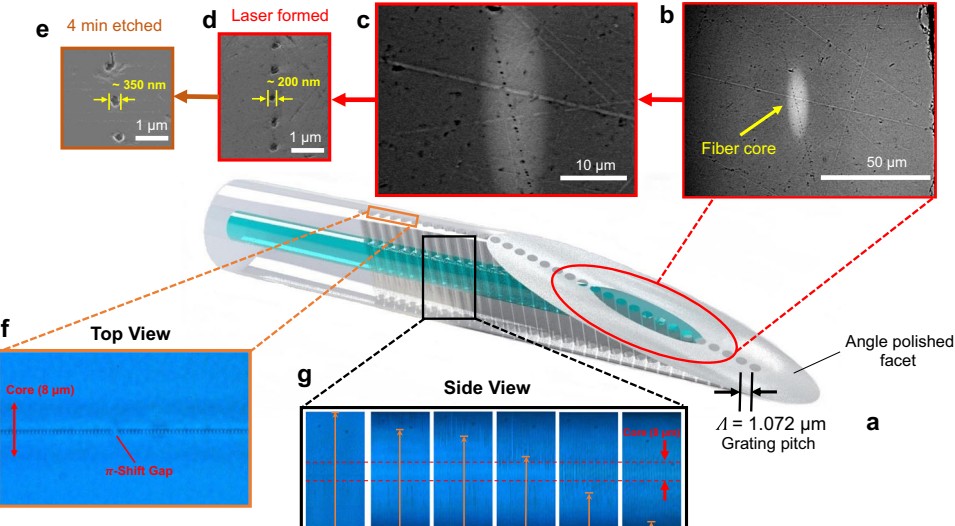

**Fig. 3 Morphological characterisation of the nanoholes with SEM and optical microscopy images.** Schematical view of a nanohole embedded optical fibre with angled facet (**a**) and supporting microscopic imaging (**b**–**g**). Optical microscopy of the nanohole array showing the central alignment of the array to the core in the end view (**f**) and a time sequence (left to right) of evaporating isopropanol with menisci level (yellow arrow) moving downward in side views of the nanoholes (**g**). Under increasing magnification, SEM images of the angled fibre facet (**a**–**d**) unveil open nanoholes having formed continuously and uniformly (~200 nm diameter) through the core (white oval zone) and cladding without a breakthrough on the tight 1.072 μm periodic spacing. SEM (**e**) shows the hole diameter expanded to 350 nm after 4 min of chemical etching (5% HF).

time sequence of microscope images showing the evaporation of isopropanol from fully wetted (i.e., air-filled) nanoholes, recorded over ~10 s (see also Supplementary Movie 1, 2). A wide range of solvent types was found to readily wet and fill the nanoholes of such laser structured fibre.

The provision of RI-matching oil around the fibre cladding (Fig. 1) did not inhibit near-neighbour filament nor nanohole formation. Lateral damage zones were insignificant both at surfaces and the internal volume such that isolated holes could be formed, pulse by pulse, in arrays having periods as small as 1.072 μm (Fig. 3f, g). The point-by-point writing permitted the insertion of optical defects such as the π-shift identified in the optical microscope image (Fig. 3f). The end-view of the nano-channel array, formed on 1.072 μm periodicity confirmed a sub-micron lateral resolution of hole positions inside the core waveguide, and extended well beyond the beam depth of focus (1.4 μm) to reach through the fibre cladding cross-section (Fig. 3g).

The cross-sectional morphology of the nanohole structures was further examined by scanning electron microscope (SEM) imaging. A section of nanohole FBG was cleaved and polished at an oblique angle (~30°), as shown schematically in Fig. 3a. Electron microscopy of the oblique fibre cross-section identifies an array of nanoholes having formed through the core and cladding (Fig. 3b). With an increasing magnification of the core section, the sequence of SEM images (Fig. 3b–e) confirm the formation of isolated nanoholes having a relatively uniform cylindrical shape, and positioned on $\Lambda = 1.072$ μm period with an estimated hole diameter of ~200 nm. A slightly irregular hole perimeter is noted with an internal surface roughness of ~50 nm, which is ~30× smaller than the optical probe wavelength. Hence, the holes can form fully isolated in tightly packed arrays having nearly uniform cross-section through the core (Fig. 3) and a majority of the cladding. The nanohole arrays are only noted to break into each at the cladding surface over a depth of ~15 μm. Hence, nanoholes with a high-aspect ratio >500× have been demonstrated in the fibre. The present results have not unveiled the high-stress zones expected to be forming around the nanoholes as seen previously[42,49] in bulk glass when formed

with pulses of longer duration (1 ps). Such densified zones may form a barrier to micro-crack formation to toughen the glass in the same way chemical treatment stressing works in display glass. A tensile proofing test of mechanically stripped fibre with and without the nanohole array was completed[50,51], providing the cumulative failure probability for fracture as shown in Supplementary Fig. 2. As expected, there was only a modest decrease in the strength of the nano-structured fibre, falling to ~30% of the level for mechanically stripped fibre.

Emersion of the FGBs in diluted hydrofluoric (HF) acid (5%) provided a second processing step to reproducibly increase the hole diameter, for example, to ~350 nm diameter after 4 min (Fig. 3e). Such diameter tuning opens a means for tuning the photonic stopband response, as well as increasing the FBG sensitivity to RI changes.

**Photonic stopband response.** The validation of strong and responsive photonic stopbands is presented in the reflection spectra of Fig. 4a for nanohole arrays filled with air and a range of solvents (Table 1). Under high RI contrast with air ($n_H = 1.0$) or Oil10 ($n_H = 1.66$), the nanohole array provided broad (~2 nm) and strong (~3 dB) Bragg resonances with only 600 holes (i.e., 643 μm length). The evolution of the spectral profile, reflection peak and linewidth with the increasing grating length for the case of air-filled holes (Supplementary Fig. 3) followed the growth trends of traditional FBGs, expect to develop much more rapidly owing to a ~100-fold higher RI contrast ($\Delta n = 0.45$) over traditional FBGs ($\Delta n \sim 10^{-3}$)[2]. Although the 200 nm hole diameter encompassed only a ~2% overlap with the modal field (MFD $\cong$ 10.4 μm), the gratings provided a strong effective coupling strength of up to $\kappa_{AC} = 9.17\ \mathrm{cm}^{-1}$. The strong responses are enabled by the strong field repulsion or attraction effects of the guided light into or out of the planar grating zone (Fig. 2b versus Fig. 2c) for the cases of low and high RI, respectively. As a result, the Bragg resonance had shifted definitively from 1549.93 nm in the air ($n_H = 1.0$) to 1551.76 nm in Oil10 ($n_H = 1.661$), providing an average RI sensitivity of 2.75 nm/RIU across a wide RI-sensing range. The progression of strong-to-weak stopbands, from

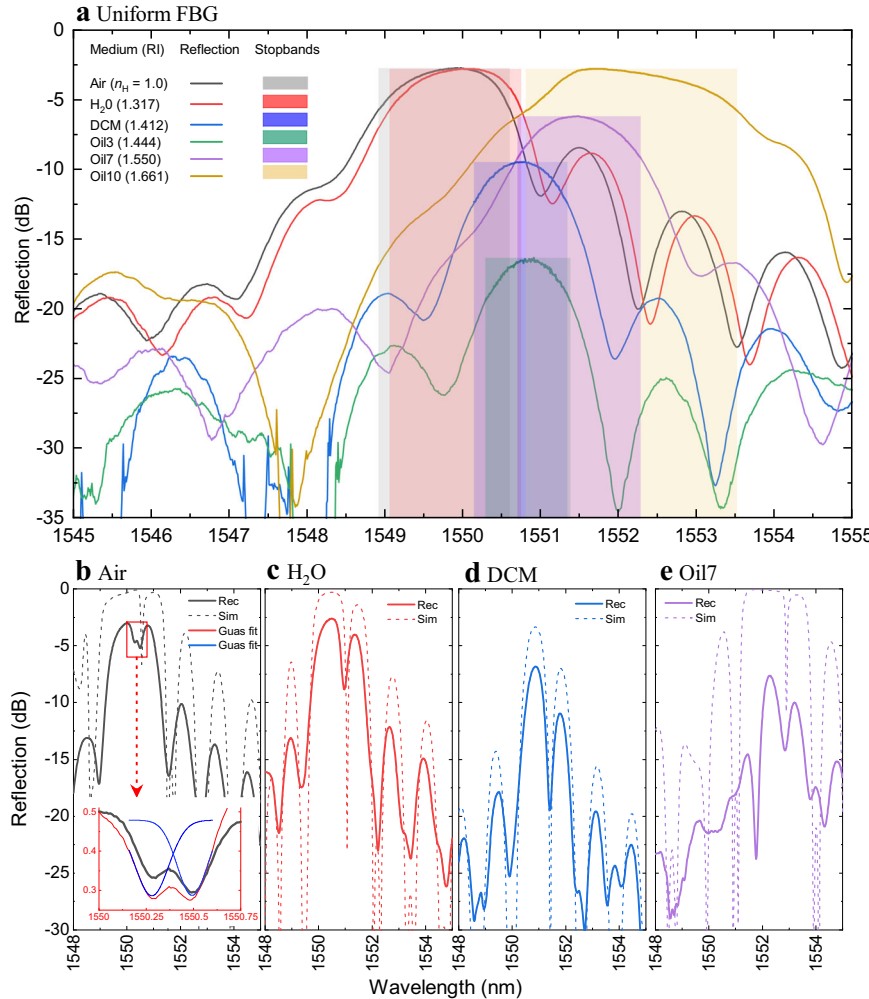

**Fig. 4 Photonic stopband response of the FBGs filled with different materials.** Reflection spectra (**a**) recorded from a uniform FBG of 600 nanoholes ($E_{pulse}$ = 4.5 μJ, $\Lambda$ = 1072 nm, 2 mm plate) forming strong (3 dB) and wide (~2 nm) stopbands when filled with low ($n_H$ = 1) or high ($n_H$ = 1.661) refractive index materials. The coloured bands mark the 3 dB bandwidth. Refractive index matching oil ($n_H$ = 1.448) dramatically weakens the stopband. Reflection spectra (solid lines) of unpolarized light (**b–e**) were recorded from a π-shifted FBG of 1200 nanoholes for four different examples of RI values ($n_H$ by colour from legend **a**) and comparison with simulated spectra for P-polarised light (dashed lines). Laser-induced birefringence of δλ = 210 pm is resolved for the air-filled case (**b** inset). Gaussian line fits of the π-defects (blue and red lines in **b** inset) provide ±10 pm spectral precision.

−2.72 dB reflection with air to −17.28 dB with Oil5 ($n_H$ = 1.448), marks the matching condition on RI, beyond which higher RI (i.e., Oil10, $n_H$ = 1.661, Fig. 4a) regenerated strong stopbands (−2.76 dB).

Although strong in reflection, the breadth of the stopbands under high RI contrast is limiting in specifying the precise centre Bragg wavelength. A more rewarding direction was obtained by implanting a π-defect into a Bragg grating array of 1200 nanoholes, opening well-resolved passbands (Fig. 4b–e, Supplementary Fig. 7). In the progression for nanoholes filled with air, water, dichloromethane and Oil7 ($n_H$ = 1.0, 1.316, 1.412, and 1.550 in Fig. 4b–e, respectively), the π-defect provided the highest contrast (>10 dB) and sharpest resolution ($\delta\lambda_B$ = 150 pm) pass-bandwidth (3 dB) when tuned nearer to the refracting index matching condition, $n_H \cong n_G$ (Fig. 4d and see also Supplementary Fig. 7a). For air-filled nanoholes, the resonance is separated into two ~3 dB peaks ($\delta\lambda_B$ = 210 pm, Fig. 4b), unveiling a waveguide birefringence of $\Delta n_H$ = 0.07. Otherwise, the birefringence was unresolved and served only to broaden the passband ($\delta\lambda_B$ = 150–400 pm) with increasing RI contrast examined in the range 1.31 < $n_H$ < 1.67 (Supplementary Fig. 8a). The point-by-point writing of such high-aspect ratio nanoholes with <100 nm

positional precision thus facilitated photonic band shaping with sharp spectral features tuned to ~100 pm resolution.

EME modelling, (Fig. 4b–e, dashed line spectra) provided a good representation of the π-shifted grating reflection (Fig. 4b–e, solid line spectra) with the hole diameter optimised to ~200 ± 10 nm for the 1200 nanohole array design ($\Lambda$ = 1.072 μm). The simulated spectra have reproduced the broad stopband and sideband features as well as the narrow π-defect response and accurately tracked the wavelength shifts for the full range of RI changes from $n_H$ = 1.0 (Fig. 4b) to 1.550 (Fig. 4e). However, the grating reflection fell short of the simulated peak values by 3 dB to 7 dB over the respective air ($n_H$ = 1.0) to Oil7 ($n_H$ = 1.550) cases, pointing to unaccounted losses from Rayleigh scattering on nanohole surface roughness (~50 nm, Fig. 3), variances of up to ±50 nm on the nanohole positioning, and first-order Bragg radiation.

To improve on the resolution for RI sensing, Gaussian-shaped functions were found to best match the π-defect transmission dip, for example, as shown in the case of the air-filled holes (Fig. 4b inset). The spectral matching enabled centre resonant wavelengths to be specified with a precision of ±10 pm. Otherwise, the Gaussian representations furnished linewidths varying from 180

**Table 1 Refractive index (RI) of liquids applied inside of nanohole FBGs.**

| Medium | Abbreviation | RI at $\lambda = 1550$ nm[55] |
|---|---|---|
| Air | Air | 1.002 |
| Water | $H_2O$ | 1.3164 |
| Methanol | MeOH | 1.3174 |
| Acetone | Ace | 1.3483 |
| Isopropanol | IPA | 1.3737 |
| Dichloromethane | DCM | 1.4124 |
| Cargile AA-1.450 | Oil1 | 1.440 |
| Cargile AA-1.452 | Oil2 | 1.442 |
| Cargile AA-1.454 | Oil3 | 1.444 |
| Cargile AA-1.456 | Oil4 | 1.446 |
| Cargile AA-1.458 | Oil5 | 1.448 |
| Cargile A-1.540 | Oil6 | 1.522 |
| Cargile A-1.572 | Oil7 | 1.550 |
| Cargile A-1.586 | Oil8 | 1.562 |
| Cargile A-1.630 | Oil9 | 1.600 |
| Cargile B-1.700 | Oil10 | 1.661 |
| Cargile M-1.705 | Oil11 | 1.670 |

to 360 pm (FWHM) for the unresolved birefringence cases (Fig. 4c–e; see also Supplementary Fig. 7).

**Photonic bandgap engineering—etching**. The as-formed nanoholes provided added utility in guiding chemical etchant to tune the hole diameters and further engineer the stopband response. SEM revealed a near-linear response of increasing hole diameter with etching time, beginning from the ~200 nm diameter for the laser-formed nanohole and expanding to 350 nm for the case of 4 min etching in 5% HF acid (Fig. 3e). In real-time monitoring of FBG reflection spectra (1200 holes), the stopbands in both uniform and π-shifted gratings (Supplementary Fig. 4a) shifted monotonically to a shorter wavelength over a 7 min etching time. The blue spectral shifting arises from a drop in effective RI as the lower RI of etchant ($n_H \cong n_{H2O} \cong 1.3164$) displaces the higher RI glass ($n_G = 1.45$). Increasing hole diameter is noted to weaken and narrow the stopbands (Supplementary Fig. 4b) from −3.7 to −29 dB in strength and 1.5–0.97 nm in bandwidth, over the 7 min etching time. However, the π-shifted passband (not plotted) retained a full 9 dB contrast inside of the FBG stopband, even as the stopband weakened by > 20 dB over the full 7-min etching time (Supplementary Fig. 4b). The preservation of sharply resolved π-defect resonances attests to a highly ordered and unbroken patterning of the high-aspect ratio nanoholes, that does not breakthrough on a tight packing density (1.072 μm period) even as diameters were opened up over the 7 min etching time.

Nanohole FBGs with uniform gratings were prepared with different etching times (0, 2, 4, and 6 min) to provide a widely varying FBG response (Supplementary Fig. 5) when filled with solvents or oils spanning a large range of RI values (Table 1). The nanohole arrays as formed by the laser (0 min) typically offered the strongest stopbands (Fig. 5a). The progression to weaker stopbands (Fig. 5a–c) varied from strongly to weakly for solvents having the lowest (Fig. 5a, $n_H = 1$), matched (Fig. 5b, $n_H = 1.448$) and highest (Fig. 5c, $n_H = 1.6$) values of RI. This progression is noted in the plot of peak Bragg reflectance (Fig. 5d) over the full range of tested RI values (Table 1). The fall-off was most pronounced for air ($n_H = 1$), decreasing by >25 dB over the 6 min etching time. The fall-off was delayed in the case of higher RI oils with RI values of $n_H = 1.522–1.66$, where reflectivity first increased to a maximum of ~5 dB for 2–4 min etching time. In all cases, the stopbands were weakest for the 6 min etching time. The bandwidth (3 dB) of the Bragg stopbands was also strongly influenced by the chemical etching time, either narrowing by ~50% or broadening by more than threefold according to the

negative ($n_H < 1.45$) or positive ($n_H > 1.45$) contrast of RI (Fig. 5e).

The alignment of the EME-modelled spectra (Supplementary Fig. 1) with the observed reflection spectra (Fig. 4, Supplementary Fig. 5) provided a precise, semi-empirical estimate of the nanohole diameter, for example, yielding $500 \pm 20$ nm diameter for 4 min etching time (Supplementary Fig. 6). In this way, hole diameters of 220, 300, 500, and 700 nm were assigned with variances of ~±10 nm to FBGs opened with 0, 2, 4, and 6 min etching time, respectively. The spectral corroboration identifies a peak value of FBG reflectance (Supplementary Fig. 4, Fig. 5d) arising on the first quarter wavelength resonance of the nanohole diameter (i.e., $\lambda/4n_H$), for example, encompassing hole diameters of 220–300 nm for solvents varying from $n_H = 1$ for air to $n_H = 1.41$ for Dichloromethane. The steep fall-off of reflection thus aligns with an anti-resonance on doubling of the hole diameter to $\lambda/2n_H$, corresponding to diameters of 500–700 nm for air to high index oil ($n_H = 1$ to 1.60). Hence, the diameter of nanoholes transitions from a first-order resonance for the strongest stop reflection band (0–2 min etching) to the first anti-resonance (6 min) over the ranges of laser formation and chemical etching tested here.

An increasing hole diameter further provided stronger spectral shifts of the stopbands (Supplementary Fig. 5, Fig. 5a–c), moving in reversed directions as expected depending on the positive ($n_H > n_G$) or negative ($n_H < n_G$) contrast of solvent RI with respect to the glass index. The largest hole diameter (~700 nm for 6 min) and highest RI oils offered the highest wavelength sensitivity, with the Bragg centre wavelength shifting by up to +40 nm for $n_H = 1.60$ solvent over the 0 to 6 min etching time. In contrast, a smaller and negative wavelength shift of −2.2 nm was noted (Fig. 5a) for the case of the air-filled holes over the same 6 min etching time. Under index matching (Fig. 5b, $n_H = 1.45$), the Bragg resonance did not shift.

**High-resolution RI sensing**. In order to provide the highest resolution RIU sensing, π-shifted FBGs of 1200 hole arrays were modified with similar etching times (0, 2, 4, and 6 min). EME modelling offered close matching of the spectra strong bands, side lobes, and π-shift passbands (Fig. 4b–e), yielding similar values of effective hole diameters, corresponding to 220, 300, 500, and 700 nm for the respective 0, 2, 4, and 6 min etching times. The π-defect passbands (Supplementary Fig. 7) were spectrally fitted to Gaussian line shapes, varying from 200 to 400 nm linewidth (Supplementary Fig. 8). A narrow birefringent splitting (~200 pm) of the π-defect was occasionally resolved in the spectra (Supplementary Fig. 8) in cases with the strongest stopbands and largest contrast in the RI.

With spectral line fitting (Supplementary Fig. 7) of the π-defects, centre Bragg wavelengths could be determined to ±10 pm precision, enabling RI sensing to a high-resolution of $10^{-5}$ RIU. When plotted against the RI (Fig. 5f, solid fonts), the π-resonance shifts demonstrated an impressive RI response of FBG stopbands, shown globally over an extraordinary range of RI values ($n_H = 1–1.66$) and nanohole diameters (220–700 nm). The EME modelling (Fig. 5f, dashed line) followed each data set to ±200 pm (rms) spectral precision, relying only on one value of optimised hole diameter across the full RI testing range. Discrepancies in the air (±500 pm) and between methanol and water may arise from surface tensions effects that require further study. Sharply forming π-defect resonances were identified in all cases except the two highest indexes ($n_H = 1.661$ and 1.670) and large diameter (6 min) condition (open triangle Fig. 5f), where stopbands became overly broad and mixed with sidebands (see Supplementary Fig. 5d).

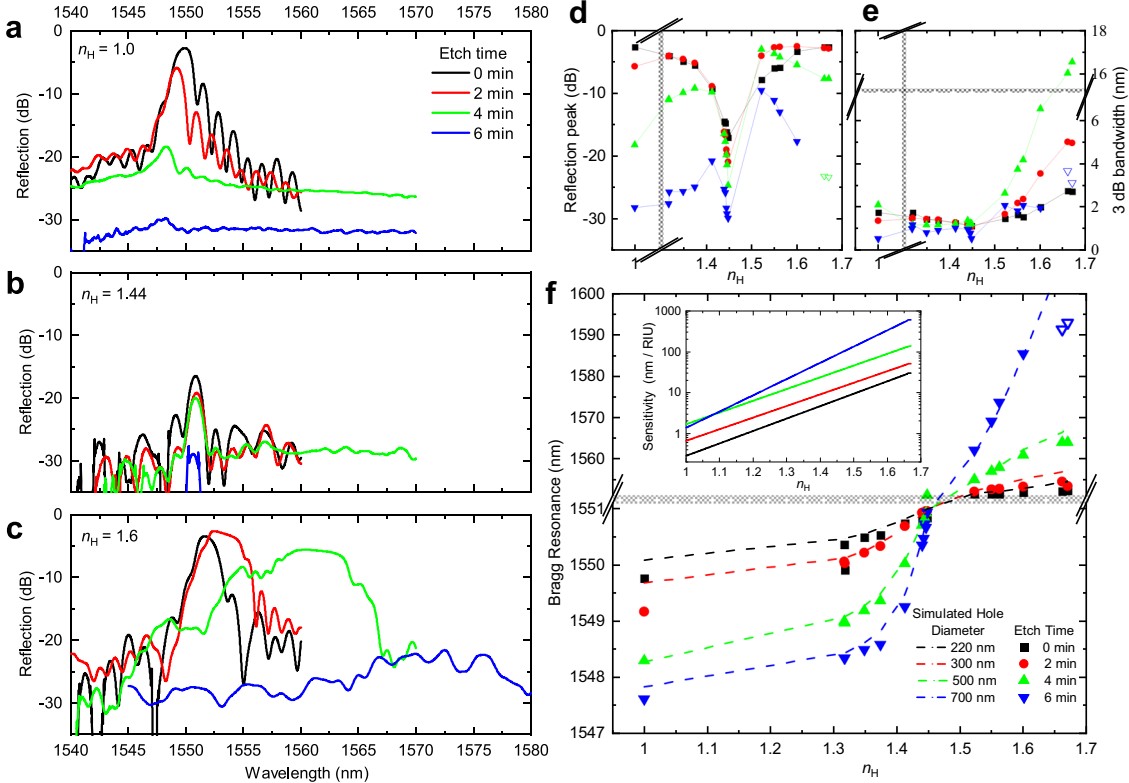

**Fig. 5 High-resolution RI sensing enabled by FBGs with different hole diameters.** Reflection spectra recorded from uniform FBGs ($E_{pulse} = 4.5\,\mu J$, $\Lambda = 1072$ nm, 600 holes) filled with low (**a** $n_H = 1$), medium (**b** $n_H = 1.44$), and high (**c** $n_H = 1.661$) refractive index materials, revealing the influence of increasing nanohole diameter due to varying chemical etching time (0–6 min). Peak reflection strength (**d**) and 3-dB linewidth (**e**) of the Bragg resonances plotted as a function of refractive index for different chemical etching times (following legend **a**). Bragg resonance wavelength (shifts on right axis) plotted versus refractive index **f** for different chemical etching times. A global fit of the data by EME simulation is presented (dashed lines) for hole diameters of 220–700 nm (see legend). The slopes of data provide a strongly increasing refractive index sensitivity (inset) for increasing RI and increasing hole diameter (colour-coded lines). Shaded zones mark axis scale changes.

The narrow π-shifted stopbands provided strong optical responses to sense the local environment, as demonstrated in the wide 1548–1590 nm shift in Bragg wavelengths (Fig. 5f). The slopes of these responses yielded a widely ranging response in RIU sensitivity (Fig. 5f, inset) that reached as high as 600 nm/RIU for the case of highest RI ($n_H = 1.66$) and largest hole diameter (700 nm). This RIU sensitivity is comparable to the best FBG-based demonstrations to date (i.e., 945 nm/RIU in ref. [52]). Moreover, spectral line fitting (±10 pm) offered a high-resolution determination of the RI to ±$10^{-4}$ precision.

The chemical etching of the filaments demonstrated an attractive progression of improving RIU sensitivity response (Fig. 5f, inset), but at the cost of forming broader and weakened stopbands as the hole, the diameter was increased from 200 nm to 700 nm (i.e., see Fig. 5 and Supplementary Fig. 5). This range of subwavelength hole diameter (i.e., below $\lambda/n = 1.072\,\mu m$) was beneficial in circumventing optical resonances in the holes that interfere with the photonic stopbands. In this way, strong stopbands enabled a signal to noise ratio (SNR) amenable for optical sensing over the moderately large range of reported RI values, 1–1.67. Scaling to a larger hole diameter would otherwise diminish the SNR and narrow the RI-sensing range, for example, falling sevenfold to the 1.32–1.41 range as reported in ref. [37] for 1.665 μm hole width. The benefit of the sub-micron hole dimensions is thus narrower and stronger stopbands that avoid ambiguity in sensing multiple Bragg resonances in distributed FBG systems. Moreover, the nanohole FBG avoids a low limit cut off in the RI (i.e., n∼1.48) for total internal reflection as in the case of evanescence-based FBG refractometry[14,29,30].

The nanohole array presents a responsive optical fibre sensor, retaining cladding strength while readily wetting with numerous solvents that can reach into the open photonic structure formed along with the fibre core. The nanoholes are amenable to hosting a wide range of materials, for example, such as nematic liquid crystals (NLC)[53]. The small, 200 nm diameter capillaries were found to impose a strong axial molecular alignment of NLC, manifesting in strong optical birefringence in the fibre polarisation modes that facilitated an all-fibre dynamically switchable polarisation filter[53]. The facile means of laser writing and patterning of high-aspect-ratio nanoholes introduced here thus opens the realm for creating strong and compact photonic stopband devices directly in traditional optical fibres while facilitating environmental sensing through a thick robust cladding. The line-by-line writing is an intrinsically fast process, being single-step and scalable to sub-second exposure times with only modest repetition rates of ∼1 kHz in comparison with current fabrication techniques[37,52]. The fabrication method is extensible to other types of fibres and fibre materials, or to the structuring of two-dimensional arrays of nanoholes in the fibre[54]. Such structuring would permit the engineering of compact 2D photonic bandgap devices directly inside of the fibre that is attractive for tailoring the coupling to cladding modes or in and out of the fibre through the radiation modes[46,54]. In this way, the minimally invasive methods of filament explosion and chemical etching are enabling in the photonic bandgap engineering of traditional optical fibre, promising to transform how fibres shape the flow of light and sense the local environment from applications in biomedical probes through to large area communication networks.

## Methods

**Laser fabrication of hollow-filament arrays**. A frequency-doubled Yb-doped fibre laser (Amplitude Systems, Satsuma) provided femtosecond pulses of 515 nm wavelength, 250 fs pulse duration and $M^2 < 1.2$ beam quality. The 800 kHz repetition rate was down counted to 1 Hz, to provide up to 8 µJ pulse energy to the filament-forming exposure arrangement (Fig. 1). The laser beam was expanded ~2 times (Linos Magnification Beam Expander, 4401-257-000-20) to fill the full aperture of an aspherical focusing lens (New Focus, 5722-A-H) of 0.55 NA. A 0.8 µm spot radius ($1/e^2$ intensity) and 1.4 µm depth of the focus is expected if focussed directly into silica glass.

In the arrangement of Fig. 1, a long and uniform filament beam shape with a diameter of <1 µm was generated by surface aberration induced by the flat surfaces of fused silica plates (Nikon, NIFS-S (S-grade)) positioned between the lens and optical fibre. Filament track lengths from 80 to 125 µm were generated with 2 and 3 mm plate thickness, respectively. The surface aberration stretched the laser energy over a large depth of focus in a similar way to an axicon forming a Bessel-like beam[42]. In this way, high pulse energy was applied without inducing distortion from Kerr lens focusing and plasma defocusing effects, while retaining the nonlinear interaction benefits of narrowing modification to a sub-diffraction-limited size.

The experimental approach was adapted from ref. [45] wherein a standard single-mode telecommunication fibre (Corning, SMF-28) was mounted in contact with the bottom surface of the aberration plates (Fig. 1). Index matching oil (Cargille, 50350) was applied to fill the gap between the fibre and plate and remove cylindrical aberration by the fibre. Laser filaments were aligned laterally to ±1 µm precision to bisect the waveguide core (Fig. 3f), and shifted vertically to illuminate any portion of core and cladding (Fig. 3g) by using a three-axis motion and alignment system (Aerotech Inc., Aerotech-PlanarDL-00 XY and ANT130-060-L Z) having 3 nm resolution. Filament positions in the fibre cross-section (Fig. 3f, g) were verified by backlighting the fibre with an optical microscope (Olympus, BX51).

The laser pulse energy was adjusted upward from the low-contrast conditions until the optical contrast of filament tracks forming in the fibre was observed to darken and indicate the onset of micro-explosion, opening hollow-filament shapes inside of the fibre. Single-pulse exposure of 3 and 7 µJ energy opened hollow filaments of 80 µm to 125 µm length, respectively, with silica plates of 2 and 3 mm thickness, respectively. The axial focusing was optimised to avoid the burning or evaporating of the RI-matching oil and minimise ablative machining and near-surface damage at the fibre cladding surface. In this way, blind (2 mm plate) or fully open (3 mm) holes (Fig. 3g) were formed in the fibre core and cladding. The filament tracks could be aligned side-by-side to ~1 µm spacing and remain isolated without breaking through or undergoing beam propagation or processing distortion by the pre-existing track. Filament tracks were assembled into a tightly packed linear array and precisely centred along with the fibre core (Fig. 3f, g) for evaluation for FBG responses.

**Nanohole morphology**. The nano-scale formation of high-aspect ratio holes into the optical fibre was examined by mechanically cleaving and optically polishing a laser-exposed fibre at an oblique angle (~30°) (Fig. 3a). A filament array of nanoholes was generated through the cladding and core waveguide regions on 1.072 µm period, using 4.5 µJ pulse energy, and 2 mm thick aberration plate. The cleaved facet was polished with silicon carbide paper (1200 grit), whereas wetted with a few drops of distilled water. After cleaning with acetone and isopropanol, the polished fibre facet (Fig. 3a) was viewed in orthogonal alignment with an SEM (Hitachi, SU5000), providing high contrast views of the full fibre cross-section, shown in Fig. 3b–d. The SEM images confirmed a ~200 nm diameter of the open nanoholes forming through a majority of the filament length, defining a high-aspect ratio of ~500 times. Hole-to-hole breakthrough was not observed over most of the filament length for periods as small as $\Lambda = 1.072$ µm, or for identical nanohole arrays opened to the larger diameter (i.e., 350 nm, Fig. 3e) after chemical etching.

**Capillary flow**. Optical microscopy was used for confirming and monitoring the wetting, capillary flow and evaporation of liquids with various values of RI (Table 1) into and out of the laser-formed nanoholes, providing optical images (Fig. 3g) and video recordings (Supplementary Movies 1 & 2). The wetting times varied from sub-seconds for solvents (i.e., water, methanol, acetone, etc.) (Merck) to tens of seconds for index matching oils (Cargile). Dichloromethane (DCM) and acetone were applied to remove oils and to clean nanoholes before immersing the fibre in a new liquid.

**FBG characterisation**. At low pulse energy exposure (~350 nJ), low-contrast filament arrays were previously shown to assemble in silica fibre on $\Lambda = 0.536$ µm period and provide strong first-order Bragg grating responses at 1550 nm[45]. However, the opening of nanoholes with the higher pulse energy (3–7 µJ) required in the present work resulted in filament distortion and breakthrough of holes when positioned on a similar first-order period. The formation of isolated nanoholes was verified by SEM (Fig. 3d) with a doubling of the period to $\Lambda = 1.072$ µm, thus enabling a second-order FBG response at 1550 nm wavelength. π-shifted FBGs

were fabricated for narrowing the spectral response of the device to below 200 pm linewidth (3 dB). FBG spectral responses were recorded real-time during the laser fabrication and chemical etching, or during the filling or evaporation of various liquids (Table 1). Reflection spectra were excited with a 1530–1610 nm wavelength broadband source (Thorlabs, ASE-FL7002), and recorded through an optical fibre circulator (Thorlabs, 6015-3-FC) by a high-resolution optical spectrum analyser (Anritsu, MS9740B). The influence of laser exposure and the number of filaments were evaluated. Supplementary Fig. 3 shows the influence and limitation of increasing device length on the reflection peak and bandwidth. A 600 and 1200 element array of nanohole filaments were selected to study the FBG responses to solvents, chemical etching, and to compare uniform and π-shift gratings.

**Chemical etching**. Femtosecond laser irradiation followed by chemical etching (FLICE)[56–58] was adopted here for opening the nanohole diameter. FLICE extended the diameter of the laser-formed nanoholes in a predictable and reproducible way. The laser-formed FBGs were immersed in acetone and followed with DCM to remove oils and debris. Fibres were submerged in 5% dilute HF acid solution for up to 7 mins. Longer etching times were found to degrade the FBG spectrum. After etching, the fibre was promptly immersed in distilled water, then IPA and left to air dry. Identical FBGs were prepared with 2, 4, and 6 min of etching and then spectrally characterised underfilling with all solvent and oil types listed in Table 1. Assessment by a combination of SEM morphology (Fig. 3e), FBG spectral responses (Fig. 5f and Supplementary Fig. 5), and simulations (Supplementary Fig. 1) provided an estimated etching rate on the nanohole diameter beginning more slowly at 40 µm per minute in the first 2 minutes and rising to a steady value of 100 nm/minute thereafter.

**Optical modelling**. Simulation of the 3D light intensity distribution (Fig. 2b, c) and the spectral reflections (Fig. 2a and Supplementary Fig. 1) expected from a nanohole array positioned in the core waveguide of SMF-28 fibre was provided by commercial software (Lumerical Inc.), based on finite difference time domain (FDTD: 3D Electromagnetic Simulator) and eigenmode expansion methods (EME; MODE: waveguide simulator), respectively.

For the FDTD simulation, intensity patterns (Fig. 2b, c) were provided for a second-order FBG of 100 nanoholes on $\Lambda = 1.072$ µm pitch and having a nanohole diameter of 300 nm. Core and cladding RI values were matched to SMF-28 fibre at 1550 nm wavelength. A perfect electrical conductor was imposed at the centre-fibre symmetry plane ($z = 0$ in Fig. 2b, c) with absorbing boundary conditions on the outside borders. A spatially uniform mesh with grid size of $\Delta x = \Delta y = \Delta z = 48$ nm was applied with time steps of $\Delta t = 0.092$ fs. Long programme running time-limited simulations to FBGs having a linear array of 100 filaments that spanned ~100 µm along with the fibre. Intensity patterns did not deviate materially when compared with the simulation of larger arrays of up to 600 nanoholes.

The EME simulation provided reflection spectra of FBGs (Fig. 2a, Fig. 4b–e, Supplementary Fig. 1, Supplementary Fig. 3, and Supplementary Fig. 6) containing arrays of 600 or 1200 nanoholes that are spaced uniformly or π-shifted, respectively. A non-uniform spatial mesh was defined as having the finest pitch for lateral dimension beginning at $\Delta y = \Delta z = 10$ nm inside the fibre core array, and gradually increasing to 400 nm in the cladding. A uniformly small pitch of $\Delta x = 6$ nm was maintained axially. A unit cell was defined around the $\Lambda = 1.072$ µm period of nanoholes and sandwiched by a pair of periodic boundary conditions. The programme provided a fundamental single-mode having a mean-field diameter of ~10 µm and returned reflection spectra through the S-parameter function.

## Data availability
The data collected and generated during the study are available from the corresponding authors upon reasonable request.

## Code availability
The codes used in the simulations are available from the corresponding authors upon reasonable request.

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

## Acknowledgements

The authors are grateful to Gligor Djogo and Dr. Stephen Ho for guidance on chemical etching and SEM imaging, to Dr. Jianzhao Li for experimental support and to Dr. Ehsan Alimohammadian for aberration tuning of filament shape. The authors are also appreciative of professor Xijia Gu, Michael Bakaic, and Nicholas Burgwin for providing fibre proof tester and their advice on fibre tensile strength measurements. This work was supported by the Natural Sciences and Engineering Research Council of Canada (NSERC, grant nos.: STPGP 521526-18, CREAT 484907-16), the Edward S. Rogers Sr. Graduate Scholarship, and Mitacs Accelerate Fellowship (IT16189).

## Author contributions

Device fabrication and characterisation were conducted by A.R. and K.M.A. E.E. provided guidance with generating laser filaments and with SEM characterisation. K.M.A. carried out the theoretical modelling. The manuscript was written by A.R., K.M.A, and P.R.H. and all authors contributed to editing. P.R.H. directed the overall project.

## Competing interests

The authors declare no competing interests.
