## [Peer Review File · Nature Communications]

REVIEWER COMMENTS

Reviewer #1 (Remarks to the Author):

In this paper, the authors present a pretty nice work on fabricating periodic arrays of tightly-packed holes chemically etched down to the fiber core, from the cladding, along a standard optical fiber for precise sensing of the refractive index of liquids.

The results are clean, well detailed and properly supported by FEM modelling.

I have a concern of novelty for publishing this work to a high impact journal such as Nature Communications, where novelty is paramount. The results can be seen as a direct extension of similar works presented way before, for example one decade ago in the cited reference 37 (Yang et al., "Rapid fabrication of microhole array structured optical fibers," Optics Letters, vol. 36, p. 3879, 10 2011). The only differentiator claimed by the authors is the reduction on the period of the inscribed FBG from 1.35 microns (in ref.37) to 1.076 microns (in the current manuscript). The authors describe in a much more detailed fashion and with a quite better control over the parameters pretty much the same experiment, for the same application of RI sensing. The authors mentioned at the end of page 1 that the larger structures reported in ref. 37 limit the RI measurement window, but this is not well supported in the manuscript. For example, in Fig.5, the authors show that increasing the hole's diameter (from 200 to 700nm) leads to an increasing sensitivity over RI.

In addition, the authors justify their technique by claiming that it leads to an improved mechanical robustness of the fiber compared to other competitive methods. There is no mention about any measurement of the fiber's mechanical strength resulting from the proposed fiber processing technique. I guess that the steps of affecting the cladding structure with laser filaments, followed by chemical etching of holes through the fiber, lead to a significant reduction of the fiber robustness. I suggest the authors properly support this aspect of their claims with quantitative data.

Reviewer #3 (Remarks to the Author):

The paper is well written and highlights some key and innovative results in the field of engineering and sensing applications.

The expected impact could be very large as highlighted in the conclusion part since such laser processing of high aspect ratio nanoholes (eventually coupled to chemical etching) opens the door toward photonic band gap engineering in 2D and even 3D by femtosecond laser direct writing. However the key innovation, as stated in the introduction of the paper, is rather related to the fiber sensing part that falls down in a very rich context.

So the introduction describes correctly the rich literature about fiber sensing and refractive index optical sensing techniques and the problem is well positioned with respect of the context. Then the paper is very technical, quite focused on optical sensing and brings two key things:

- Writing high aspect ratio nanoholes in fibers to realize and engineer photonic band gaps
- To use such laser writing to develop a FBG (Fiber Bragg gratings) based sensors with a very high sensitivity (typ 600nm/RIU) that is at the top of the current state of the art in this specific field

Note that writing high aspect ratio nanoholes in oxide glasses like silica has been already well studied, especially in bulk glasses, and maybe the author could cite early development like the work of F. Courvoisier group and not only the two recent papers from R. Stoian and F. Courvoisier:

<https://aip.scitation.org/doi/abs/10.1063/1.3479419>

<https://www.osapublishing.org/ol/abstract.cfm?uri=ol-34-20-3163>

<https://www.osapublishing.org/oe/fulltext.cfm?uri=oe-18-2-566&id=194323>

So the innovation in this "laser manufacturing" section relies on a high quality technical realization related to the femtosecond laser processing of optical fibers to realize an array of such high aspect ratio nanoholes assembly to realize an engineered photonics band gap based on FBG design. This is quite challenging and the authors successfully imprint FBG with 200nm holes, 1micron pitch and even Pi phase shift. The realization is really convincing and supported by optical modeling and characterization of the photonics stop band response accompanied with some SEM and optical observations.

One minor question is about these nanoholes that should exhibit a densified shell around and thus stress/strain that likely cumulates along the grating structure and might lead to cracks or fragile fibers anyway?

Then the paper focuses on a demonstration related to fiber based sensing using FBG based photonics stop band. They report some results at the best of the state of the art in terms of sensitivity and resolution with typically 600nm / RIU. Maybe here the author could remind the advantages of their approach in comparison to the other publication that reports similar quantitative results (ref 46). This part is well studied with many clear results supported again by the adequate modeling.

Reviewer #1 Comments:

In this paper, the authors present a pretty nice work on fabricating periodic arrays of tightly-packed holes chemically etched down to the fibre core, from the cladding, along a standard optical fibre for precise sensing of the refractive index of liquids. The results are clean, well detailed and properly supported by EME modelling.

Comment 1- I have a concern of novelty for publishing this work to a high impact journal such as Nature Communications, where novelty is paramount. The results can be seen as a direct extension of similar works presented way before, for example one decade ago in the cited reference 37 (Yang et al., “Rapid fabrication of microhole array structured optical fibres,” Optics Letters, vol. 36, p. 3879, 10 2011). The only differentiator claimed by the authors is the reduction on the period of the inscribed FBG from 1.35 microns (in ref.37) to 1.076 microns (in the current manuscript). The authors describe in a much more detailed fashion and with a quite better control over the parameters pretty much the same experiment, for the same application of RI sensing.

Response:

Indeed, the paper by Yang et al. [Ref. 37] is seminal on embedding micro-hole arrays at the core waveguide of a fibre, enabling refractive index sensing on the FBG resonance. However, there are distinct differences in the methodology, advances to the nanoscale, and benefits in the photonic stopbands in the present work that reach a considerable new milestone. The advances by our paper over this prior work can be summarized as:

1. The beam delivery methods are different where Yang et al. [Ref. 37] rely on optical interference from a phase mask to generate an array of laser interaction zones separated on 1.665 μm periodicity. Generally, interference-based fabrication techniques such as based on phase masks (Ref. 37) are well developed today for large-scale FBG production, but typically require time-consuming exposures and are further limited in flexibility for tailoring or tuning of spectral features. On the other hand, the present work relies on point-by-point writing, using a single filament laser beam shape to write one phase element of the grating at a time. Point-by-point writing is considered more challenging in the field, but offers wide latitude for varying periodicity, chirping, apodization or introducing defects to tailor the spectral response in high resolution. This novelty was highlighted by Reviewer #3 in Comments 1 and 2.
2. The fundamental laser interaction physics follow different paths, wherein Yang et al. [Ref. 37] have combined phase masks with a cylindrical lens to drive self-focussing effects. The overall interaction is a gentle material modification process of assembly nanograting volumes over long exposure times. A second step of processing was then necessary to harness selective chemical etching along the nanograting volumes to open micro-sized holes at the core. In the present work, a simple aberration effect was applied to elongate the laser focal volume into a long and uniform filament shape. The nano-hole was opened by a more aggressive process of nano-explosion, physically moving material out of the exposure zone to leave a nano-hole with uniform diameter stretching over long

lengths, and reaching fully through the fibre core and cladding. This is the first time that nano-filament explosion was successfully demonstrated in optical fibre.

3. The line-by-line writing in the present paper is intrinsically a fast writing processing, being single step and scalable to sub-second exposure times with only modest repetition rates of ~ 1 kHz for example. The combined laser exposure and chemical writing steps in Ref. 37 require two processing steps, including >30 min etching time.
4. The single pulse interaction in the present paper enables near-instantaneous opening of holes that are robust and well formed with 200 nm diameters. In contrast, the chemical etching process is not nano-scale, leading to tapering of structures outside of the laser modification zones. As a result, the nanoholes in the present work are $8\times$ smaller in diameter, $17\times$ smaller in cross-sectional area and 1.55 times smaller on period. (Note that the reviewer had erroneously noted a 1.25 factor reduction by inferring a grating period of $1.35\ \mu\text{m}$ instead of $1.665\ \mu\text{m}$ in Ref. 37). This periodicity scaling was also highlighted as a novel contribution of our paper by Reviewer #3 (Comment 2).
5. The scaling from micro-scale in Ref. 37 to nanoscale in the present paper unlocks a key requirement for building strong and controllable PC stop bands. The micro-channels in Ref. 37 of $1.35\ \mu\text{m}$ diameter are larger than the wavelength in the medium, $\lambda_{\text{prob}/n_{\text{eff}}} \sim 1.07\ \mu\text{m}$, whereas the nanofilaments of $0.2\ \mu\text{m}$ diameter are only a small fraction ($\sim 20\%$) of the wavelength ($\lambda_{\text{prob}/n_{\text{eff}}} \sim 1.07\ \mu\text{m}$). Interference resonances in the larger micro-holes diminish the influence of photonic stop bands, but are not permitted with the present sub-wavelength nano-hole structures. The evidence for these effects bares out in the chemical etching of the filaments, showing a progression of improving RIU sensitivity response, but at the cost of forming broader but weakening stopbands as hole diameter was increased from 200 nm to 700 nm (i.e., see Fig. 5 and Fig. S5 in present manuscript). The larger diameter holes lead to reduced signal contrast, diminishing the available signal to noise ratio (SNR). Hence, an improved SNR with the small nano-scale holes in the present paper was instrumental in widening the range of refractive index values amenable to sensing, now expanded 7 fold from 1.32-1.41 in Ref. 37 to $1 - 1.67$ in the present manuscript.
6. Several key aspects on practical application of the new processes introduced here emerge from the point-by-point fabrication technique and nano-scale hole size: (1) the Bragg resonance is readily tunable unlike with the fixed period in the phase-mask approach of Ref. [37]; (2) wider latitude for scaling to lower and more efficient Bragg orders or shorter wavelength operation; and (3) facile shaping of the stopbands, spectral chirping or introduction of π -shifts. Further, the present fibre laser processing is robust, minimally modifying the fibre to retain strength (validated by proofing test; see response to Comment 2 below) while providing strong stopbands that can be flexibly tuned. Hence, one can anticipate a wide audience of interest on exploiting the techniques here to craft new devices or enable commercial products, that otherwise are very limiting when based on the techniques in the background work (i.e., Ref. 37).

7. The points above are also valid in comparisons with the related work in Ref. [33] on the ion-milling process and Ref. [34-36] on other femtosecond machining processes in tapered, nano, and micro fibres.

These points of novelty are have been stated in different wording in the introduction section of the manuscript, which culminate into a summary of four key novelty factors in the 3rd to 6th paragraph (lines 52 to 93). Our thought is that the current introduction has provided a sufficiently strong statement of novelty. Many of these novelty points were highlighted by Reviewer 3 as noted in Comments 1, 2, and 3. The comparison of our work to the prior publications as elaborated above on the smaller technical points are going to be a distraction from the bigger picture of novelty as presently cited. Therefore, no additional novelty statements were deemed necessary.

Comment 2-The authors mentioned at the end of page 1 that the larger structures reported in ref. 37 limit the RI measurement window, but this is not well supported in the manuscript. For example, in Fig.5, the authors show that increasing the hole's diameter (from 200 to 700nm) leads to an increasing sensitivity over RI.

Response:

We agree with the reviewer that the limitations of the weakened bands as reported for holes with large diameter was not clearly stated in the paper.

The original manuscript had presented the following ideas: The second paragraph in the “*Photonic-bandgap engineering – etching*” section (lines 171-211) described the trade off of increasing Bragg wavelength shift at the cost of weakening stopbands as hole diameter of the nano-holes increased. This manifested in the experimental data of Fig. 5 and Fig. S5 where FBGs with holes of larger diameters had generated larger wavelength shifts, but at the cost of much broader stopbands. When scaling the holes to larger diameter, namely, 200-700 nm, a higher sensitivity was indeed observed. In this range, the hole sizes were still submicron, and the Bragg wavelength liquids could be recorded accurately over a broader range of low to high values of refractive index (RI).

In comparison with Ref. 37, the substantially larger micron sized holes (1.665 $\mu\text{m} \times 3.53 \mu\text{m}$) had broadened the stopband significantly, weakening the reflectance signal significantly to limit the RI sensitivity to a much narrower (7 fold) range of RI values. The only reason a large sensing range was possible in the present work, was owing to the micro-to-nano down scaling on hole diameter.

To connect the cause and effect more clearly, the follow revisions and additions were made in lines 235-244 of the revised manuscript:

“The chemical etching of the filaments demonstrated an attractive progression of improving RIU sensitivity response (Fig. 5f, inset), but at the cost of forming broader and weakened stopbands as the hole diameter was increased from 200 nm to 700 nm (i.e., see Fig. 5 and Fig. S5). This range of subwavelength hole diameter (i.e., below $\lambda/n = 1.072 \mu\text{m}$) was beneficial in

circumventing optical resonances in the holes that interfere with the photonic stop bands. In this way, strong stopbands enabled a signal to noise ratio (SNR) amenable for optical sensing over the moderately large range of reported refractive index values, 1 - 1.67. Scaling to larger hole diameter would otherwise diminish the SNR and narrow the RI sensing range, for example, falling 7-fold to the 1.32 - 1.41 range as reported in [37] for 1.665 μm hole width. The benefit of the sub-micron hole dimensions are thus narrower and stronger stopbands that avoid ambiguity in sensing multiple Bragg resonances in distributed FBG systems. Moreover, the nanohole FBG avoids a low limit cut off in the RI (i.e., $n \sim 1.48$) for total internal reflection as in the case of evanescence-based FBG refractometry [14, 29, 30].”

Comment 3- In addition, the authors justify their technique by claiming that it leads to an improved mechanical robustness of the fibre compared to other competitive methods. There is no mention about any measurement of the fibre’s mechanical strength resulting from the proposed fibre processing technique. I guess that the steps of affecting the cladding structure with laser filaments, followed by chemical etching of holes through the fibre, lead to a significant reduction of the fibre robustness. I suggest the authors properly support this aspect of their claims with quantitative data.

Response:

Our claim had been based on day-to-day fibre handling experiences in the lab, so we agree with the reviewer that a qualitative assessment would be beneficial to the manuscript. To support our claims, a tensile proofing test of mechanical strength was applied to mechanically striped SMF-28 fibre that were either unmodified or embedded with nano-structured gratings. The new data show that there was only a modest fall in the strength of the nano-structured fibre compared with the unmodified fibre, falling 3-fold fall in strength. While we have not proof tested other forms of fibre cladding modification, our experience in handling fibres with overly modified cladding structure seen in Ref. 37 have led to exceptional fragile fibres that were difficult to handle in the lab in contrast with the present nano-filament gratings.

To reflect on these benefits, the following additional data collection and editing changes were made:

1. Proofing test data for modified and unmodified fibre were collected with a commercial fibre proof tester (Vytran PTR-100) and assessed by the standard Weibull statistics for cumulated failure probability. A new supplemental section (S2: Fibre proof test for mechanical tensile strength with embedded nano-holes) with a figure on cumulative failure (Fig. S2) and a description of the measurement procedure (Fig. S2 caption) were added as replicated here:

Newly added supplementary figure (Fig. S.2) to the revised paper.

Fig. S2 Caption :

“Fig. S2. Weibull plot showing cumulative failure probability of tensile strength comparing mechanically stripped SMF-28 fibres embedded with (red triangle) and without (blue circle) a nano-hole array of filament grating. Ten samples have been tested for each fibre type. The FBGs were fabricated with exposures parameters of $E_{\text{pulse}} = 4.5 \mu\text{J}$, $\Lambda = 1072 \text{ nm}$, and 600 holes, providing strong Bragg resonances of $\sim -2.5 \text{ dB}$ in reflection (i.e., Fig. S3) for blind holes penetrating $>70 \%$ of the cladding cross-section with diameters of $\sim 200 \text{ nm}$. The mechanical stripping induced damage to cladding as noted by breaking stress values varying over a wide range (0.6 to 1.46) with a median value of 0.95 GPa compared with a pristine fibre having $\sim 5.3 \text{ GPa}$ [49]. Fibres with the nanoholes failed withing in a narrow stressing zone of 0.24 to 0.36 GPa, with a median breaking stress of 0.33 GPa lying at approximately one-third of the breaking level for the unmodified fibre. The nanoholes therefore showed relative robustness for a highly structure fibre core and cladding. The data were collected using a commercial fibre proof tester (Vytran, PTR-100). The cumulative failure probability was calculated as a function of stress following the method used in [50].”

2. The proofing tests in new Supplement 2 were addressed in the manuscript with the following new sentences added to the Results, at the end of the Nano-hole array FBG section (lines 126-129): “A tensile proofing test of mechanically stripped fibre with and without the nano-hole array was completed [49, 50], providing the cumulative failure probability for fracture as shown in Fig. S2. As expected, there was only a modest decrease in the strength of the nano-structured fibre, falling to $\sim 30\%$ of the level for mechanically stripped fibre.”

3. Two new references pointing to similar methods of fibre proof testing have been cited in line 127 of the main text and also in the new Supplemental 2 section: [49] M. Bernier et al., *Opt. Lett.* 39, 3646 (2014); [50] X. Gu et al., *IEEE Sensors Journal*, 6, 668 (2006).
4. In further consideration of the fibre strength, the high stress zones observed by G. Zhang et al. (*Photon. Res.* 7, 806 (2019)) and T. Chen et al. (*Micromachines*, 11, 671 (2020)) to be forming around nano-holes in bulk silica glass may serve as a barrier to micro-crack formation that is not been previously cited in prior studies. Therefore, a new sentence has been added to manuscript (lines 123 to 126) to acknowledge the possible presence of similar high stress zones in the present gratings and their possible influence on the strength of the fibre, as follows: “The present results have not unveiled the high stress zones expected to be forming around the nano-holes as seen previously [42, 51] in bulk glass when formed with pulses of longer duration (1 ps). Such densified zones may form a barrier to micro-crack formation to toughen the glass in the same way chemical treatment stressing works in display glass.”
5. A new reference relating to point 4 has been added as citation in line 124: [51] G. Zhang et al. *Photon. Res.* 7, 806 (2019).

While we cannot test the same version of fibre structures formed by Sun [37] or other groups [52], our experience in previous research on forming larger cavities or holes in fiber cladding ([56] M. Haque, and P. R. Herman, *Laser Photonics Rev.* 9, 656 (2015)) have found the fiber structures to be exceptional fragile and readily prone to breakage in comparison with the present nano-hole structures. Further, to the best of our knowledge, no prior study has presented a nano-hole array sensory fibre without destroying a large part of the fibre cladding. Our expectation is that the micro-structured fibres in related literature [add relevant reference] will be significantly more fragile and yield a wide variance in breaking points that entails a study that goes beyond the scope of the present work. Therefore, the additional materials on proofing tests are an optimistic addition to the paper that show nano-structured fibre can be fabricated with potentially high strength, sufficient for opening the field for nano-structuring of fibre.

Reviewer #3 Comments:

Comment 1- The paper is well written and highlights some key and innovative results in the field of engineering and sensing applications.

The expected impact could be very large as highlight in the conclusion part since such laser processing of high aspect ratio nanoholes (eventually couple to chemical etching) open the door toward photonic band gap engineering in 2D and even 3D by femtosecond laser direct writing. However the key innovation, as stated in the introduction of the paper, is rather related to the fibre sensing part that fall down in a very rich context.

So the introduction describes correctly the rich literature about fibre sensing and refractive index optical sensing techniques and the problem is well positioned with respect of the context. Then the paper is very technical, quite focuses on optical sensing and brings two key things:

- Writing high aspect ratio nanoholes in fibres to realize and engineered photonics band gap
- To use such laser writing to develop a FBG (Fibre Brag gratings) based sensors with a very

high sensitivity (type 600nm/.RIU) that is at the top of the current state of the art in this specific field

Note that writing high aspect ratio nanoholes in oxide glasses like silica has been already well studied, especially in bulk glasses, and maybe the author could cite early development like the work of F. Courvoisier group and not only the two recent papers from R. Stoian and F. Courvoisier:

<https://aip.scitation.org/doi/abs/10.1063/1.3479419>

<https://www.osapublishing.org/ol/abstract.cfm?uri=ol-34-20-3163>

<https://www.osapublishing.org/oe/fulltext.cfm?uri=oe-18-2-566&id=194323>

Response:

We are pleased by the confirmation of novelty in our manuscript. We further agree with the comment to include the additional papers as representing the pioneers of forming nanoholes with axicon optics. The three recommended references were cited at line 54, adding [38] Bhuyan et al. APL 97, 081102 (2010); [39] Bhuyan et al. Opt. Express 18, 566 (2010); and [40] Courvoisier et al. Opt. Express 34, 3163 (2009) to the existing citations related to the same topic [41-43].

Comment 2-So the innovation in this “laser manufacturing” section relies on a high quality technical realization related to the femtosecond laser processing of optical fibres to realize an array of such high aspect ratio nanoholes assembly to realize and engineered photonics band gap based on FBG design. This is quite challenging and the authors successfully imprint FBG with 200nm holes, 1 micron pitch and even Pi phase shift. The realization is really convincing and supported by optical modeling and characterization of the photonics stop band response accompanied with some SEM and optical observations.

One minor question is about these nanoholes that should exhibit a densified shell around and thus stress/strain that likely cumulates along the grating structure and might leads to cracks or fragile fibres anyway?

Response:

While we were aware of these densified zones observed previously, i.e., in G. Zhang et al. Photon. Res. 7, 806 (2019) and T. Chen et al. Micromachines, 11, 671 (2020), their presence around the nano-holes in our work could not be confirmed by the polishing and SEM analysis (Fig. 3). One possible cause may be our significantly shorter pulse duration or differences in methodology. This densified glass was very interesting to our group as a means of compressive strain that may toughen the glass as used in display glass by chemical treatment.

We have addressed the reviewer question with the following revision to the manuscript:

1. A new sentence has been added to manuscript (lines 123 to 126) to acknowledge the expectation for such high stress zones, and their potential connection for toughening of the glass, as follows: “The present results have not unveiled the high stress zones expected to be forming around the nano-holes as seen previously [42, 51] in bulk glass when formed with pulses of longer duration (1 ps). Such densified zones may form a

barrier to micro-crack formation to toughen the glass in the same way chemical treatment stressing works in display glass.”.

2. The additional sentences cite a new reference added as follows: [51] G. Zhang et al. *Photon. Res.* 7, 806 (2019).

Comment 3-Then the paper focus on a demonstration related to fibre based sensing using FBG based photonics stop band. They report some results at the best of the state of the art in terms of sensitivity and resolution with typically 600nm / RIU. Maybe here the author could remind the advantages of their approach in comparison to the other publication that reports similar quantitative results (ref 46). This part is well studied with many clear results supported again by the adequate modeling.

Response:

We are pleased by the confirmation of the state-of-the-art results. We agree that our manuscript had skirted over a deeper discussion of comparisons with related literature which would assist with highlighting the advantages and any shortcomings of the hollow-filament fibre gratings. The advantages of our paper over prior works (e.g., Ref. 52) includes the following points:

1. In the background literature, the micro fabrication processes that were applied to create the hero demonstrations were tedious and multi-step, and were highly invasive, produce fragile components (e.g., micro-hole structuring in fibres using phase masks [37], and micro-slot fabrication in the fibre [52]). We have pointed out the advantages of our technique over these shortcomings (see our response to first comment of reviewer #1). Our expectation is that other micro-structured fibres are much more fragile than nano-hole fibres presented in our manuscript. The mechanical strength of the nano-hole fibres was a question raised by the first reviewer, which has been addressed by the proofing tests as described in our response to Comment 3 above.
2. The direct interaction of the light with the liquids in the nano-holes of our waveguiding device had provided a strong RI sensitivity over a broad range of refractive index values, which was not available in other technique such as micro structured holes (Ref. 37) or other evanescent wave based refractometers (Ref. [14, 29, 30]).
3. A further advantage of the current laser writing is the potential for scaling up to two-dimensional patterning of nano-holes arrays, opening the possibility for broader applications based on 2D or 3D photonic crystal engineering in the core of fibre.

We have modified the manuscript to expand on our technique advantages and provided a clearer statement delineating the future potential of our fabrication technique as follows:

1. Four sentences has been added to the manuscript (lines 238 to 244) to address the wider RI sensitivity available in this work to other fibre based refractometers. In particular, the influence of the submicron dimensions of the nano holes is stated as follows: “In this way, strong stopbands enabled a signal to noise ratio (SNR) amenable for optical sensing

over the moderately large range of reported refractive index values, 1 - 1.67. Scaling to larger hole diameter would otherwise diminish the SNR and narrow the RI sensing range, for example, falling 7-fold to the 1.32 - 1.41 range as reported in [37] for 1.665 μm hole width. The benefit of the sub-micron hole dimensions are thus narrower and stronger stopbands that avoid ambiguity in sensing multiple Bragg resonances in distributed FBG systems. Moreover, the nanohole FBG avoids a low limit cut off in the RI (i.e., $n \sim 1.48$) for total internal reflection as in the case of evanescence-based FBG refractometry [14, 29, 30].”.

2. The potential for expanding the range of materials amenable for filling of the nano-hole gratings was expanded to include our recent work on liquid crystals (lines 246 to 250) as follows: “ The nano-holes are amenable to hosting a wide range of materials, for example, such as nematic liquid crystals (LC) [53]. The small, 200 nm diameter capillaries were found to impose a strong axial molecular alignment of nematic liquid crystals (NLC), manifesting in strong optical birefringence in the fibre polarization modes that facilitated an all-fibre dynamically switchable polarization filter [53].”.
3. A new sentence has been added to the manuscript (lines 252 to 254) comparing the fabrication speed of the present work to other similar works as follows: “The line-by-line writing is an intrinsically fast process, being single step and scalable to sub-second exposure times with only modest repetition rates of ~ 1 kHz in comparison with current fabrication techniques [37, 52].” .
4. A couple of sentences on the advantages of 2D filament writing and their future directions was added to the manuscript (lines 254 to 257): “The fabrication method is extensible to other types of fibres and fibre materials, or to the structuring of two-dimensional arrays of nanoholes in the fibre [54]. Such structuring would permit the engineering of compact 2D photonic bandgap devices directly inside of the fibre that is attractive for tailoring the coupling to cladding modes or in and out of the fibre through the radiation modes [46, 54].”.
5. A new reference has been added to the manuscript (line 247) on adding nematic liquid crystal to the laser nano-holes as follows: [53] A. Rahnama, et. al *Advanced Optical Materials*, 2100054 (2021).
6. A new reference has been added to the manuscript (line 255) on two-dimensional patterning of nano-holes as follows: [54] A. Rahnama. et. al. *Proc. SPIE 11676, Frontiers in Ultrafast Optics: Biomedical, Scientific, and Industrial Applications XXI*, (2021).

Extra Corrections:

1. A new reference has been added to the paper (line 327) on our group background work on Femtosecond laser irradiation followed by chemical etching (FLICE) as follows: [57]: G. Djogo et al., *International Journal of Extreme Manufacturing*, 1, 11 (2019).
2. A typo in the text has been corrected in line 270 as follows: “cylindrical aberration” term was replaced by “surface aberration”.

3. A typo in Fig. 3 has been corrected as follows: the grating pitch had been mistakenly shown in the figure to be 1.076 μm and was corrected to match with the text value of 1.072 μm .

REVIEWERS' COMMENTS

Reviewer #1 (Remarks to the Author):

The authors carefully addressed my concerns about the initial manuscript and proposed a revised version of the manuscript that is now acceptable for publication from my point of view. I therefore recommend its acceptance.